# Diabetic Mastopathy. Review of Diagnostic Methods and Therapeutic Options

**DOI:** 10.3390/ijerph19010448

**Published:** 2021-12-31

**Authors:** Paweł Guzik, Tomasz Gęca, Paweł Topolewski, Magdalena Harpula, Wojciech Pirowski, Krzysztof Koziełek, Marcin Żmuda, Marcin Śniadecki, Tomasz Góra, Paweł Basta, Artur Czekierdowski

**Affiliations:** 1Department of Gynecology and Obstetrics, City Hospital, 35-241 Rzeszow, Poland; harpula.magda@gmail.com (M.H.); minddin@gmail.com (T.G.); 2Department of Obstetrics and Pathology of Pregnancy, Medical University of Lublin Poland, 20-059 Lublin, Poland; tomasz.geca@umlub.pl; 3Department of Obstetrics, Gynecology, Gynecologic Oncology and Gynecologic Endocrinology, Medical University of Gdańsk, 81-214 Gdansk, Poland; ptopolewski@gumed.edu.pl (P.T.); marcin.sniadecki@gumed.edu.pl (M.Ś.); 4Department of General Surgery, City Hospital, 34-700 Rabka-Zdrój, Poland; wpirowski@gmail.com; 5Breast diagnostic unit, St. John Paul II Hospital of HCP Medical Center in Poznań, 61-485 Poznan, Poland; k.kozielek@gmail.com; 6Clinical Department of Pathomorphology, Provincial Clinical Hospital No. 2, 35-301 Rzeszow, Poland; martinuz@tlen.pl; 7Chair of Gynecology and Obstetrics, Department of Gynecology and Oncology, Jagiellonian University Medical College, Jagiellonian University, 31-008 Cracow, Poland; pbasta@gin.cm-uj.krakow.pl; 81st Department of Gynecological Oncology and Gynecology, Medical University of Lublin, 20-059 Lublin, Poland; arturczekierdowski@umlub.pl

**Keywords:** diabetic mastopathy, breast ultrasonography, breast surgery, diabetes in breast, breast degeneration, lymphocytic mastopathy, breast tumor

## Abstract

Diabetic mastopathy is a rare breast condition that may occur in insulin-treated men and women of any age. The etiology is still unclear; however, the autoimmunological background of the disease is highly suspected. The changes in diabetic mastopathy may mimic breast cancer; therefore, its diagnostic process is demanding, and treatment options are not clear and limited. Lesions in DM are usually multiple; therefore, surgical removal is not fully effective. A well-done anamnesis with core-needle biopsy is essential and definitive in most cases. In this review, we summarize up-to-date knowledge of diagnostic methods and therapeutic options for diabetic mastopathy treatment and present three cases of diabetic mastopathy-type lesions in ultrasound and radiological examinations.

## 1. Introduction

Diabetic mastopathy (DM), also known as “lymphatic mastopathy”, “fibrocystic mastopathy”, and “fibrocystic breast degeneration”, accounts for less than 1% of all benign breast lesions [1]. The disease occurs mainly in young and middle-aged women with long-standing type I diabetes; the incidence ranges from 0.6% to 13% [1]. Cases of DM have also been described in patients with type II diabetes, autoimmune diseases such as Hashimoto’s thyroiditis, as well as in healthy individuals and in men. The condition is characterized by foci of dense fibrosis, lobular atrophy, and clusters of lymphocytes with periventricular and peritubular distribution [2]. Despite a considerable increase in our knowledge on DM that was gained in recent years and several new hypotheses, the pathogenesis of this disease remains unclear [3]. The main differential diagnosis is breast cancer because of similar clinical symptoms and imaging features [3]. Diagnosis is complex, and biopsy of the lesion is often necessary to establish a proper lesion classification.

## 2. Etiology and Pathogenesis

The exact etiology of DM is still unclear. It is generally believed to be associated with autoimmunity. It has been shown that breast lesions contain B-cell lymphocytic that infiltrate with an expression of HLA-DR antigen in the epithelium, mainly HLA-DR3, 4, or 5 positive [4]. These immunological features are reminiscent of the changes occurring in the endocrine glands in diseases such as Sjogren’s syndrome, parotitis, Hashimoto’s thyroiditis [5]. Tomaszewski et al. [6] hypothesized that fibroinflammatory changes found in the breast are partly related to hyperglycemia, which causes an expansion of the extracellular matrix followed by increased collagen production and decreased collagen degradation. According to their model, proteins undergoing non-enzymatic glycosylation act as antigens, inducing autoimmune B-lymphocyte proliferation and autoantibody production. The release of cytokines causes swelling of epithelial matrix cells and the formation of specific epithelial cells embedded in a dense fibrous stroma, epithelial fibroblasts (EFB). Similar changes are observed in other autoimmune disorders [6]. An alternative hypothesis presented by Seidman et al. suggests that DM is relatively specific to patients with insulin-dependent diabetes mellitus and may result from the body’s inflammatory response to exogenous insulin [7]. Miura et al. described the case of a patient whose serum anti-insulin autoantibodies were found to cross-react with breast milk duct epithelial cells. It has been suggested that anti-insulin antibodies produced in diabetic patients may cause ductal inflammation [2]. Although there have been numerous hypotheses for the reasons of diabetes-associated mastopathy, several cases of the disease in non-diabetic individuals have also been described, which calls into question its specificity for this type of condition and raises discussion and controversy related to the nomenclature of the disease [1,8].

## 3. Gross Pathology

The lesions typically are 0.5 to 6.0 cm in size. Most specimens do not contain a visible tumor, but a distinct firm or hard mass is palpable, and the area of involvement has a firm edge when dissected. The cut surface of the lesion appears as homogeneous white to pale-gray tissue that may be trabeculated but is often visibly indistinguishable from the surrounding fibrous breast parenchyma. Cysts and other gross alterations of proliferative breast disease are not integral parts of DM [9].

## 4. Microscopic Pathology

Chronic inflammation is seen as dense periductal, perilobular, and perivascular lymphocytic infiltrates (Figure 1). Lymphocytes are predominantly B cells intermingled with a smaller population of T cells [10].

Very few plasma cells or other leukocytes are present in the perivascular infiltrates. Germinal centers are rarely formed. Infarcts, fat necrosis, duct stasis, arthritis, and other inflammatory lesions are not considered to be characteristic features of DM [9].

The basement membrane of ducts and lobules may be markedly thickened. Lobules are small and sparse in number [10].

The lesions typically contain collagenous stroma with keloidal collagen and an increased concentration of stromal spindle cells when compared to the surrounding breast tissue [10]. Typical stromal changes include the proliferation of fibroblasts and myofibroblasts and may be due to abnormal deposition of collagen. Stroma is very dense and paucicellular, often glassy in appearance. Sometimes fibroblasts may present with enlarged nuclei and appear epithelioid in shape. Additionally, polygonal epithelioid cells may be found dispersed in the collagen among spindle cells [9,10]. Multinucleated stromal giant cells and mitotic activity typically are not present [9].

In the majority of DM cases, all of the foregoing histologic features are found, but on occasion, one or more of the typical findings may be absent [9].

From a microscopic pathology perspective, the small B-cell-rich nature of the lymphocytic infiltrates of diabetic mastopathy biopsies can mimic low-grade B-cell lymphomas, especially MALT lymphomas. This diagnostic feature may be misleading; thus, correlation with the clinical picture of the pathology is crucial.

## 5. Clinical Picture, Diagnostic Criteria

DM occurs predominantly in young to middle-aged (20–40 years old) women [11], although rarely, it may also occur in men [8,12]. In most cases, it is diagnosed in patients with type I diabetes. These patients often develop other diabetic complications, especially retinopathy, nephropathy, and neuropathy [13]. The condition can also occur in patients with type II diabetes [11] and in non-diabetic individuals. Almost all women are premenopausal when the breast lesion biopsy is performed. The interval between the onset of diabetes and the detection of the breast lesion is, on average, about 20 years. Bilateral lesions have been diagnosed in nearly 50% of cases [9], and they are frequently found in patients with various autoimmunological diseases. The number and size of lesions found in cases with diabetic breast mastopathy correlate with the progression of the underlying disease.

Clinically, the condition is characterized by breast lesions that are painless, hard (typically harder than invasive carcinoma) and easily movable on palpation, irregular, and poorly demarcated. They may occur as single or as multiple lesions, may be unilateral or bilateral, and involve all quadrants of the breast. It has been shown that most cases of DM occur in the upper lateral/medial part of the breast (76%), and their size varies from 0.5 to 3.7 cm [3]. To date, in reported cases, axillary adenopathy was not present [14,15].

Various proposed diagnostic criteria for DM have emerged over the years. Longman and Hoffman were the first to propose that the diagnosis of the disease should meet the following conditions: (1) a long history of insulin-dependent diabetes mellitus; (2) painless, hard, irregular, poorly demarcated, and mobile breast lesions that are often bilateral or unilateral; (3) on radiography-mammography, dense glandular tissue; (4) on ultrasound (US), acoustic shadows behind the lesion; (5) on fine-needle biopsy, strong resistance to puncture of the lesion [16]. Further criteria for DM were proposed by Camuto et al. [17]. They have presented the following features necessary to diagnose DM: (1) the condition affects premenopausal women with long-term type 1 diabetes mellitus, which is usually associated with vascular complications; (2) there is a palpable breast lesion that is firm, non-painful, and clinically suspicious for cancer; (3) mammography shows increased density but does not confirm the presence of a localized mass, and ultrasound also fails to identify a solid or cystic mass; (4) surgical or thick-needle biopsy of the lesion shows foci of fibrosis associated with perivascular lymphocytic infiltration. Tomaszewski et al. [6] have suggested that the diagnosis of mastopathy should consist of the following criteria: (1) lymphocytic lobular and ductal inflammation with glandular atrophy; (2) lymphatic⁄ mononuclear perivascular inflammatory infiltrates, mainly of B cells; (3) dense, often keloid-like fibrosis; (4) presence of epithelial fibroblasts (EFB). Despite various clinical criteria proposed to date, the diagnosis of DM is not unequivocal and often requires further diagnostic workup due to its high similarity to breast cancer in clinical presentation and at imaging studies [3].

## 6. Imaging Findings

Imaging findings may mimic carcinoma or be nonspecific [10]. A typical mammogram reveals localized increased density or a heterogeneous parenchymal pattern, but no radiographic features have been specifically associated with this condition. Most of the reported cases describe DM mammograms as ill-defined solid lesions, asymmetric densities, or architectural distortions with features highly suggestive of malignancy [18]. In some cases, the mammographic appearance of the mass resembles carcinoma or a fibroadenoma [10].

At ultrasound examination, DM cases are typically presenting as an ill-defined hypoechogenic mass with posterior acoustic shadowing without visible vascularity and no flow at color Doppler imaging [14]. Early DM stages may also present as well-circumscribed lesions and hypoechogenic masses without any marked acoustic shadowing [18].

Contrast-enhanced MRI may be used for the differentiation of DM and breast malignancy [14]. DM-type lesions present a homogeneously low enhancement with a gradual and progressive course and subsequent contrast wash-out seen at MRI imaging [13,19]. Despite suspected in DM low enhancement of the mass that is explained by high dense fibrous tissue content with low cellularity, hyperintensive lesions may also be found in cases with marked lymphocytic infiltration [20]. Unfortunately, in such cases, DM is strongly mimicking breast malignancy, and it may be impossible to effectively differentiate the type of the lesion at imaging studies alone.

In diffusion-weighted magnetic resonance imaging (DWI), apparent diffusion coefficient (ADC) values are used to differentiate between malignant and being breast tumors [20,21,22]. DM lesions are suspected of presenting lower signal intensity and higher ADC values on DWI as they both depend on tumor cellularity.

We present two cases of DM found at imaging studies and confirmed with histopathological examination (Figure 2, Figure 3). Three additional ultrasound examinations of young patients with DM are attached in Appendix A.

## 7. Prognosis and Treatment

Aggressive treatment is not required for the management of asymptomatic DM; however, if minor symptoms are present, painkillers are recommended. In such cases, education of the patients is crucial to reduce their fear of breast cancer diagnosis, and an annually performed mammography is recommended [23].

Fine-needle biopsy is not recommended due to the presence of increased fibrosis. Core-needle biopsy remains to be a golden standard [24,25] in all reported cases of DM suspicion. The sample of the lesion provides sufficient material for the definitive histological diagnosis. The risk of complications is low, and, in particular, no major deformations of the breast following this kind of biopsy are expected.

In difficult cases of suspected DM, clinicians should use complementary imaging methods such as ultrasound, mammography, and MRI. It is important to remember that even “overdiagnosed” DM may still result in better long-term health effects than the missed diagnosis of early breast cancer. Although core-needle biopsy is a recommended method for the preliminary diagnosis of breast tumors such as DM, the results of microscopic examination may remain inconclusive; therefore, an appropriate correlation between histopathological diagnosis with imagining tests is crucial [26].

Although DM lesions are benign, it is not unusual that new masses will develop and will be found in the same patient at follow-up imaging exams. Excision of the whole lesion may be necessary for the final evaluation if, on core biopsy, the findings are not definitive [10]; however, a high recurrence rate (30%) of DM masses was reported [8,14]. Surgical excision with an adequate normal breast tissue margin remains the primary treatment option. It was also suggested that wider margins during the excision may decrease the recurrence rate [27]. Re-excision should be avoided if recurrence in patients with prior DM history is found [1].

Other non-invasive treatment modalities are very limited to up to date. There are no reported, effective treatment methods, mostly due to fibroid-like tumor internal structure.

Each diagnosed case of DM should be managed individually. The historical flowchart in cases of diagnosed mass or DM recommended surgical excision of the tumor. In patients presenting with morphological features of DM at ultrasound or mammographic imaging, observation was recommended [1]. Type of surgery (mastectomy with immediate reconstruction or lumpectomy) should be carefully planned to achieve the best life quality and optimal aesthetic effect. The surgical treatment plan should also be evaluated according to tumor and breast size [1], number of lesions, patient’s expectations, family history, and experience of the surgical team.

Although spontaneous regression and clinical disappearance of DM have been reported, women with DM can also develop mammary carcinoma, and, therefore, any mass that is found in these women should be subjected to careful diagnostic assessment [9]. However, there is no evidence to suggest that DM predisposes to the development of mammary carcinoma or stromal neoplastic diseases such as fibromatosis [9].

To date, only one case report was published and described breast cancer found in a 48-year-old patient with DM [28], and because of this rare incidence, mastectomy is not recommended procedure in surgical management. In only a few reported cases, subcutaneous mastectomy was performed due to increased pain in the breast.

The long-term prognosis in this condition is still unknown, however as it is a benign breast tumor, most of the afflictions (aesthetic, palpable) may result from the tumor’s size.

## 8. Conclusions

Because of overlapping with breast cancer features as seen on ultrasound or mammographic imaging, DM remains a difficult diagnostic problem that may lead to unnecessary invasive procedures. In order to correctly diagnose this condition and not to miss a breast malignancy, multimodal imaging is usually indicated. Although DM is a rare clinical entity, due to increasing clinical consciousness, the number of patients with this condition is expected to grow in the near future. We believe that patients presenting DM risk factors deserve special attention during the diagnostic process, and breast cancer exclusion should remain the highest priority in differential diagnosis.

Despite many new cases presented since 1984, i.e., when the disease was first described [29], multiple deficiencies in understanding DM pathogenesis still remain unanswered. To date, no widely accepted diagnostic guidelines of DM exist, and the introduction of meaningful prognostic models requires further investigation.

It is important to correlate clinical, imaging, and pathological findings and propose both diagnosis and treatment plans by multidisciplinary teams to provide the best health care options to the patient. If a conservative approach is selected, the crucial issue is to educate the patient on how to perform breast self-exam correctly as well as to develop a plan of control visits and imaging studies in order not to miss an accidental breast cancer if it develops in the affected breast. Since current management of this condition is controversial, more awareness regarding DM differential diagnosis is necessary.

## 9. Clinical Key Points

DM is a rare condition occurring in patients with diabetes mellitus history (both types 1 and 2), may coexist with other diabetic complications such as retinopathy, neuropathy;The condition is benign, may affect one or both breasts, be localized in any breast quadrant, may be any size, typically no axillary adenopathy is found;Since the differential diagnosis of DM may be difficult, it is important to use multimodal imaging studies, i.e., sonography, mammography, and/or MRI DWI, to estimate the risk of malignant breast lesion;Core-needle biopsy and histopathological examination should be performed in all cases for the differential diagnosis;In histologically confirmed DM lesions that produce no symptoms conservative approach and observation may be offered to affected women;In symptomatic cases, surgery is a treatment of choice; however, the decision to operate and the type of operation (mastectomy with immediate reconstruction vs. lumpectomy) should be carefully planned; however, our experience shows as excisional treatments are barely needed;Approximately 30% of cases will recur, and performing wider excision margins may decrease the recurrence rate.

## Figures and Tables

**Figure 1 ijerph-19-00448-f001:**
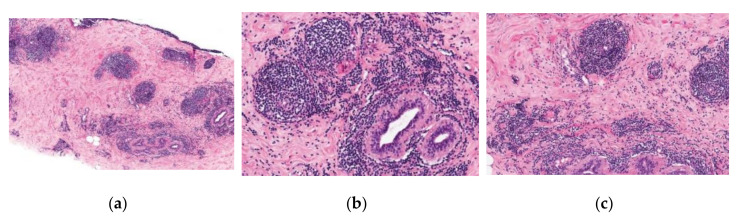
Microscopic examination of core-needle breast biopsy sample: (**a**–**c**) dense periductal, perilobular and perivascular chronic inflammatory reaction. Hematoxylin and eosin staining. Photos from Rosen’s Breast Pathology, 5th Edition.

**Figure 2 ijerph-19-00448-f002:**
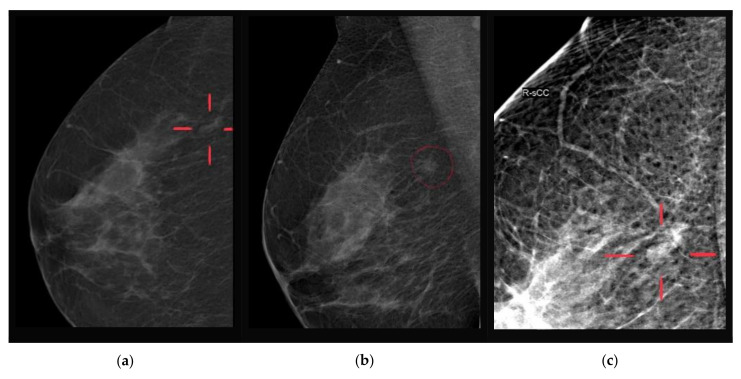
Screening mammograms of a 55-year-old female during insulin therapy, no breast symptoms, negative breast cancer family history, diabetic mastopathy confirmed microscopically following core-needle biopsy: (**a**) RCC view, 6 mm asymmetrical density of the right breast, (**b**) RMLO view, 6 mm asymmetrical density of the right breast, (**c**) “spot view”, atrophy of the ducts, periductal lymphocyte infiltration.

**Figure 3 ijerph-19-00448-f003:**
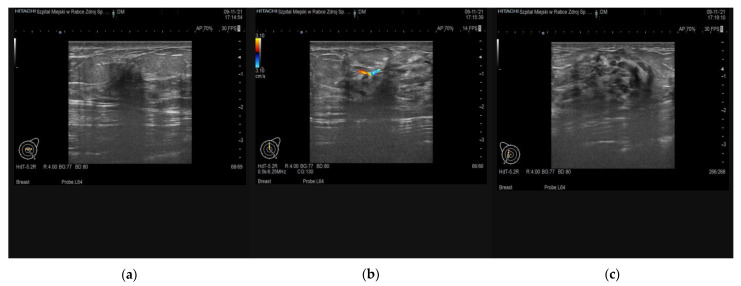
B-mode ultrasound of a 41-year-old female with type I diabetes mellitus and Hashimoto disease, positive family history of melanoma, diabetic mastopathy, and following core-needle biopsy: (**a**) 16 × 14 mm irregular, hypoechogenic lesion with blurred/spicular margins, BI-RADS-US class 4c (**b**) 18 × 6 mm irregular, hypoechogenic lesion with blurred/spicular margins, visible vascularity on color Doppler, BI-RADS-US class 4c (**c**) 25.7 × 20 mm oval shape breast lesion with parallel to the skin orientation, well-defined margins, heterogeneous echogenic structure and acoustic shadowing, BI-RADS-US class 4a.

## Data Availability

All of used data is available publicly at Pubmed (https://pubmed.ncbi.nlm.nih.gov accessed on 11 November 2021).

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
