# Peer review of "Diabetic Mastopathy. Review of Diagnostic Methods and Therapeutic Options"

_ijerph, 2021, doi:10.3390/ijerph19010448_

Round 1
Reviewer 1 Report
Congratulation on discussing a rare and interesting issue of diabetic mastopathy. You have presented it in an effective and detailed way and updated the current status of knowledge regarding this subject. My only concern is corrected with references mentioned in the article. After updating them will recommend the work for publication.
Author Response
Dear reviewer,
Thank you for your exact examination and review of our work. We appreciate a lot your effort to make our article better. We corrected and updated references in our article.
Yours sincerely,
T. Gęca MD, PhD
P. Guzik MD, PhD
PS
Merry Christmas !!!
Reviewer 2 Report
The small B-cell rich nature of the lymphocytic infiltrates of diabetic mastopathy biopsies can mimic low-grade B-cell lymphomas, especially MALT lymphomas from the pathology perspective. Adding information regarding this diagnostic consideration may be useful to at least a portion of those referencing your nice review of this topic.
Author Response
Dear reviewer,
Thank you for your exact examination and review of our work. We appreciate a lot your effort to make our article better. Agreeing with your suggestion we added: „From microscopic pathology perspective the small B-cell rich nature of the lymphocytic infiltrates of diabetic mastopathy biopsies can mimic low-grade B-cell lymphomas, especially MALT lymphomas. This diagnostic feature may be misleading, thus correlation with clinical picture of the pathology is crucial.” as the last two sentences of the „Microscopic pathology paragraph”.
Merry Christmas !!!
Yours sincerely,
T. Gęca MD, PhD
P. Guzik MD, PhD
Reviewer 3 Report
I miss more extended description of MR imaging findings and challenges in the article. Please enter an image.
Dynamic MR imaging and DWI MR must be subscribed.
Discussion on when performing a mammography is usefull.
Abstract:
DM are usual multiple, therefore.... in most of the cases
Imaging Findings
ImaginG
Despite suspected in DM..... explanation is not correct, moreover you jump from MR to hypoechogenic. This must be rewritten.
Author Response
Dear reviewer,Thank you for your exact examination and review of our work. We appreciate a lot your effort to make our article better. Following your suggestion we changed text in:
- Abstract- from primary „Lesions in DM are usually multiple, since surgical removal is not fully effective. A well-done anamnesis with core-needle biopsy is essential and definitive in most of cases.” to „Lesions in DM are usually multiple, therefore surgical removal is not fully effective. A well-done anamnesis and core-needle biopsy is essential and definitive in most of the the cases.”
- Imaging Findings, according to your suggestion we rewritten sentences „Despite suspected in DM low enhancement of the mass that is explained by high dense fibrous tissue content, hyperechogenic lesions may also be found in cases with marked lymphocytic infiltration [20].” to „Despite suspected in DM low enhancement of the mass that is explained by high dense fibrous tissue content with low cellularity, hyperintensive lesions may also be found in cases with marked lymphocytic infiltration [20].”
- Imaging Findings- we added extended description with pathological explanation of DM lesions in DWI: „In diffusion-weighted magnetic resonance imaging (DWI) apparent diffusion coefficient (ADC) values is used to differentiate between malignant and being breast tumors [20, 21, 22]. DM lesions are suspected to present lower signal intensity and higher ADC values on DWI as they both depend on tumor cellularity.”
Unfortunately, non of our patients underwent contrast enhanced MRI or DWI, thus we cannot subscribe images from these types of examination.
Yours sincerely, T.Gęca, MD, PhD
P. Guzik MD, PhD PS Merry Chistmas !!!